# Proteomic Profiling of Plasma- and Gut-Derived Extracellular Vesicles in Obesity

**DOI:** 10.3390/nu16050736

**Published:** 2024-03-04

**Authors:** Pedro Baptista Pereira, Estefania Torrejón, Inês Ferreira, Ana Sofia Carvalho, Akiko Teshima, Inês Sousa-Lima, Hans Christian Beck, Bruno Costa-Silva, Rune Matthiesen, Maria Paula Macedo, Rita Machado de Oliveira

**Affiliations:** 1Metabolic Diseases Research Group, iNOVA4Health, NOVA Medical School, Faculdade de Ciências Médicas, Universidade NOVA de Lisboa, 1169-056 Lisboa, Portugal; pedro.baptista.pereira@edu.nms.unl.pt (P.B.P.); estefania.torrejon@nms.unl.pt (E.T.); ines.ferreira@eithealth.eu (I.F.); akiko.teshima@nms.unl.pt (A.T.); ines.lima@nms.unl.pt (I.S.-L.); 2Computational and Experimental Biology Group, iNOVA4Health, NOVA Medical School, Faculdade de Ciências Médicas, Universidade NOVA de Lisboa, 1169-056 Lisboa, Portugal; ana.carvalho@nms.unl.pt (A.S.C.); rune.matthiesen@nms.unl.pt (R.M.); 3Centre for Clinical Proteomics, Department of Clinical Biochemistry, Odense University Hospital, DK-5000 Odense, Denmark; hans.christian.beck@rsyd.dk; 4Champalimaud Physiology and Cancer Programme, Champalimaud Foundation, 1400-038 Lisboa, Portugal; bruno.costa-silva@research.fchampalimaud.org

**Keywords:** extracellular vesicles, obesity, prediabetes, metabolism, proteomics, post-translational modifications, protein acetylation, protein glycation

## Abstract

Obesity entails metabolic alterations across multiple organs, highlighting the role of inter-organ communication in its pathogenesis. Extracellular vesicles (EVs) are communication agents in physiological and pathological conditions, and although they have been associated with obesity comorbidities, their protein cargo in this context remains largely unknown. To decipher the messages encapsulated in EVs, we isolated plasma-derived EVs from a diet-induced obese murine model. Obese plasma EVs exhibited a decline in protein diversity while control EVs revealed significant enrichment in protein-folding functions, highlighting the importance of proper folding in maintaining metabolic homeostasis. Previously, we revealed that gut-derived EVs’ proteome holds particular significance in obesity. Here, we compared plasma and gut EVs and identified four proteins exclusively present in the control state of both EVs, revealing the potential for a non-invasive assessment of gut health by analyzing blood-derived EVs. Given the relevance of post-translational modifications (PTMs), we observed a shift in chromatin-related proteins from glycation to acetylation in obese gut EVs, suggesting a regulatory mechanism targeting DNA transcription during obesity. This study provides valuable insights into novel roles of EVs and protein PTMs in the intricate mechanisms underlying obesity, shedding light on potential biomarkers and pathways for future research.

## 1. Introduction

Obesity is a growing global concern, with over 1 billion people affected, and unfortunately, the numbers continue to rise [1]. While genetics play a role, the increasing global prevalence of obesity is partially due to environmental factors, such as dietary habits, which are major contributors to the obesity epidemic [2,3,4]. The adoption of a Westernized lifestyle, which promotes greater consumption of calorie-rich and palatable foods while reducing physical activity, is a root cause of the obesity epidemic. For instance, overconsumption of foods with over 30% of total daily energy intake from fat, has contributed to the increase in obesity in the past years [3,4,5]. Additionally, the consumption of high-fat foods is strongly associated with a pro-inflammatory state, leading to deleterious consequences for organismal homeostasis [6]. Furthermore, excessive consumption of certain carbohydrates, particularly rapidly digestible carbohydrates, has been linked to suboptimal glycemic control in individuals with diabetes [7]. Therefore, obesity arises from an imbalance between energy intake and expenditure, resulting in excess energy being stored as fat in various organs, especially in adipose tissue [8]. This excess fat is associated with systemic complications, including hypertension, dyslipidemia, insulin resistance, and type 2 diabetes (T2D), which pose a significant challenge to healthcare [9,10,11].

Obesity impacts all organs; thus, its progression and associated comorbidities are modulated by inter-organ communication networks. Importantly, extracellular vesicles (EVs) play a crucial role in this complex inter-organ dialogue, intimately linked to the pathophysiology of obesity and diabetes [12,13]. EVs are bilayer vesicles that typically measure between 40 and 5000 nm in diameter, acting as carriers of diverse bioactive molecules, such as proteins, nucleic acids, and lipids [14,15,16,17]. Several studies have highlighted the impact of organ-derived EVs on metabolic homeostasis [16,18,19,20]. Importantly, growing evidence strongly suggests that EV-mediated crosstalk between adipose tissue, liver, and skeletal muscle is a key contributor to the development of insulin resistance [21,22,23,24,25]. Furthermore, EVs released by adipose tissue trigger abnormal activation of immune cells and endothelial dysfunction, which could explain vascular complications linked to obesity [11,26,27]. Crucially, when lean mice were administered EVs isolated from control mice but loaded with miRNAs associated with obesity, it resulted in increased glucose intolerance and hepatic steatosis [28]. Conversely, the administration of EVs from lean mice to obese mice ameliorated these conditions [29]. This highlights the significant role of miRNAs carried by EVs in triggering obesity-associated traits, even within a healthy context.

Although the miRNA content of EVs has received considerable attention, there is a knowledge gap concerning the protein cargo of these vesicles in the context of diet-induced obesity. Recently, we observed significant alterations in the protein content of gut-derived EVs isolated from a diet-induced obese mouse model [30]. The gut plays a significant role in maintaining metabolic homeostasis and is a key player in the development of obesity through various actions, including microbiota composition, hormone secretion and nutrient absorption, and its crucial connection to the central nervous system through the gut–brain axis [31,32]. Interestingly, bariatric surgery, the most effective intervention for weight loss, involves alterations in the anatomy of the small intestine and leads to T2D remission even before significant weight loss occurs [33,34]. Individuals with T2D who undergo bariatric surgery and achieve T2D resolution display distinct protein profiles in their circulating EVs, setting them apart from those in whom T2D persists post-surgery [35]. This underscores the importance of defining proteome profiles of gut EVs in the context of obesity and their potential role as predictors of resolution for obesity-related comorbidities following bariatric surgery.

Circulating EVs form a dynamic network that facilitates inter-organ communication by transporting a diverse array of organ-derived EVs, collectively forming a heterogeneous pool of EVs. These EVs act as messengers, transmitting vital molecular information across various cells and organs. It is worth noting that the number of small EVs is elevated in the circulation of patients with insulin resistance [36] and in individuals with T2D [37]. Notably, obese individuals exhibit a substantial (approximately 10-fold) increase in plasma EV levels compared to those maintaining a healthy weight [38,39,40]. On the contrary, interventions such as hypocaloric diets, exercise, and weight loss through bariatric surgery have been shown to reduce plasma EV levels [38,39,41].

Protein post-translational modifications (PTMs) play a significant role in obesity and diabetes by influencing insulin signaling and glucose metabolism, and thus, the development of associated complications [42,43,44]. Two significant post-translational modifications (PTMs) are acetylation and glycation. The first, acetylation, is closely linked to increased levels of acetyl-CoA and NAD+. The second, glycation, is connected to elevated blood glucose levels [45,46,47,48,49]. While lysine acetylation is thought to play a crucial role in maintaining energy homeostasis [50], limited reports suggest that acetylation impacts the sorting of specific proteins and/or RNA molecules into EVs [51]. Moreover, lysine acetylation is important in both immunological and metabolic pathways and helps in maintaining the equilibrium between energy storage and expenditure [52]. Glycation is a spontaneous chemical reaction between certain amino acids and reducing sugars, leading to the formation of advanced glycation end-products (AGEs) [53,54]. AGEs are considered to be the primary culprits behind various diabetic complications [55,56]. Importantly, one major precursor of AGE formation is methylglyoxal, a byproduct of glycolysis, strongly linking glycation to hyperglycemia, diabetes, and obesity [46,49]. Interestingly, glycated hemoglobin (HbA1c) is used as a marker for the diagnosis of diabetes [57,58]. Nevertheless, there is a clear need for further research delving into these PTMs in the context of metabolic diseases.

Here, we evaluate the protein content of EVs in obesity, providing crucial insights into proteomic changes and PTMs in plasma and gut EVs. This sheds light on the complex mechanisms of diet-induced obesity and prediabetes, highlighting potential biomarkers and pathways for future research.

## 2. Materials and Methods

### 2.1. Mouse Models

Six-week-old male C57Bl/6J mice were housed in a temperature-controlled room under a regular light/dark cycle of 12 h and with food and water ad libitum. For induction of prediabetes, mice were fed a high-fat diet (HFD) (OpenSource Diet, D12331) composed of 16.5% protein, 25.5% carbohydrate, and 58% fat with 13% addition of sucrose for 12 weeks. Control mice were fed a normal chow diet (NCD) (Special Diets Services, RM3) composed of 26.51% proteins, 62.14% carbohydrates, and 11.35% fat for the same period. After 12 weeks of diet, mice were subjected to an 8 h fasting period, followed by a 2 h feeding window, concluding with a 12 h fasting period. Mice were then anesthetized with isoflurane and euthanized. Blood was collected into a heparinized tube, which was then centrifuged at 500× *g* for 10 min. The supernatant was collected and centrifuged at 3000× *g* for 20 min to obtain plasma, which was stored at −80 °C for EV isolation. Three biological replicates were performed per experimental condition (diet-induced obesity and control groups). Each replica was generated by pooling plasma from 20 animals. Experimental protocols were approved by the ethics committee of the NOVA Medical School (nr.82/2019/CEFCM).

### 2.2. Intra-Peritoneal Glucose Tolerance Test (ipGTT)

In the 11th week of diet, an intra-peritoneal glucose tolerance test (ipGTT) was performed after an overnight fasting. Basal glycemia and weight were measured, and then an intra-peritoneal injection of glucose (Sigma Aldrich, St. Louis, MO, USA) at 2 g/kg body weight was administered. Blood glucose levels were measured by tail tipping at 15, 30, 60, 90 and 120 min after the injection using a OneTouch Ultra glucose meter (LifeScan Inc., Milpitas, CA, USA). The evaluation of the glycemic response was performed by calculating the total area under the whole glucose excursion, using the blood glucose concentration at timepoint 0 min as the baseline.

### 2.3. Hematoxylin–Eosin Staining

The liver was rinsed with phosphate buffered saline (PBS) and placed in 2% paraformaldehyde (PFA) with 20% sucrose overnight at 4 °C. After three PBS washes, the liver was transferred to 30% sucrose solution for 2 h at 4 °C, followed by an overnight incubation in optimal cutting temperature (OCT) compound with 30% sucrose solution (1:1 ratio). Next, the liver was embedded in molds using a mixture of OCT compound and 20% sucrose solution (3:1 ratio) and stored at −80 °C. Then, the liver was sliced into 6 μm thin sections in the cryostat and stored at −80 °C for following hematoxylin–eosin + Oil Red O staining.

### 2.4. Extracellular Vesicles Isolation

Plasma stored at −80 °C was centrifuged at 12,000× *g* for 20 min, supernatant was then subjected to ultracentrifugation (Beckman Ti70, rotor 70Ti, Brea, CA, USA) at 100,000× *g*, for 140 min. The EV-enriched pellet was collected. This pellet was resuspended in filtered PBS and layered on top of a sucrose solution. This sucrose solution was prepared with 30 g of protease-free sucrose (Sigma), 2.4 g of Tris-base (Sigma) in 100 mL of D_2_O (Sigma); pH was adjusted to 7.4. Resuspended EVs over the sucrose cushion were centrifuged at 100,000× *g*, for 70 min. The fraction of sucrose cushion containing EVs was collected and transferred to a new tube with PBS. An overnight centrifugation was then performed at 100,000× *g*, and the EV pellet was collected and resuspended in filtered PBS.

### 2.5. Nanoparticle Tracking Analysis (Nanosight)

The concentration and size of EVs were analyzed in a NanoSight NS300 (NS3000) (Malvern Panalytical, Malvern, UK) following the manufacturer’s guidelines. Briefly, EVs were diluted at a ratio of 1:1000 in filtered sterile PBS. Each sample was analyzed for 90 s, with measurements taken five times using the NanoSight automatic settings.

### 2.6. Protein Quantification

We prepared protein extracts using a lysis buffer composed of 20 mM Tris-HCl at pH 7.4, 5 mM EDTA at pH 8.0, 1% Triton-X 100, 2 mM Na_3_VO_4_, 100 mM NaF, and 10 mM Na_4_P_2_O_7_, supplemented with protease inhibitors (cOmplete^TM^, Mini, EDTA-free protein inhibitor cocktail tablets, Roche (Basel, Switzerland), Sigma). Plasma EVs were homogenized in lysis buffer and subjected to three rounds of sonication (Sonifier SFX 150, Branson, MO, USA), each lasting 10 s at 10 μm amplitude, with cooling on ice between each sonication. Lysates underwent centrifugation at 18,000× *g* for 10 min at 4 °C. The resulting supernatant was collected, and the total protein concentration was determined using the Pierce^TM^ BCA Protein Assay kit (Thermo Fisher, Waltham, MA, USA).

### 2.7. Nano-LC-MS/MS Analysis

Each biological replicate was analyzed twice, yielding two technical replicates. Peptide samples were analyzed by nano-LC-MS/MS (Dionex RSLCnano 3000, Sunnyvale, CA, USA) coupled to an Exploris 480 Orbitrap mass spectrometer (Thermo Scientific, Hemel Hempstead, UK) virtually as previously described [59]. In brief, peptide samples (5 μL) were loaded onto a custom-made fused capillary pre-column (2 cm length, 360 μm OD, 75 μm ID) packed with ReproSil Pur C18 5.0 µm resin (Dr. Maish, Ammerbuch-Entringen, Germany) with a flow of 5 μL per minute for 6 minutes. Trapped peptides were separated on a custom-made fused capillary column (25 cm length, 360 μm outer diameter, 75 μm inner diameter) packed with ReproSil Pur C18 1.9-μm resin (Dr. Maish) with a flow of 250 nL per minute using a linear gradient from 89% A (0.1% formic acid) to 32% B (0.1% formic acid in 80% acetonitrile) over 56 min.

Mass spectra were acquired in positive ion mode applying an automatic data-dependent switch between one Orbitrap survey MS scan in the mass range of 350–1200 *m*/*z* followed by higher-energy collision dissociation (HCD) fragmentation and Orbitrap detection of fragment ions with a cycle time of 2 s between each master scan. MS and MSMS maximum injection time were set to “Auto”, and HCD fragmentation in the ion routing multipole was performed at normalized collision energy of 30%, and ion selection threshold was set to 10,000 counts. Selected sequenced ions were dynamically excluded for 30 s. MS resolution was 120,000 and MS/MS resolution was 15,000.

### 2.8. Database Search

Mass accuracy was set to 5 ppm for peptides and 0.01 Da for ionized fragments. Trypsin cleavage allowing a maximum of four missed cleavages was used. Carbamidomethyl was set as a fixed modification. Methionine oxidation, lysine and N-terminal protein acetylation (Appendix A), lysine glycation (carboxymethyl, carboxyethyl, and pyrraline) (Appendix A), arginine glycation (argpyrimidine, glyoxal and methylglyoxal-derived hydroimidazolones, and tetrahydro pyrimidine), glutamine deamidation and asparagine deamidation were set as variable modifications. The MS/MS spectra were searched against all reviewed protein sequences available in a standard mouse proteome database from UniProt (UP000000589). For the search, all protein sequences were also concatenated in reverse order, with lysine and arginine residues maintained in their original positions. The data were searched and quantified with both MaxQuant [60] and VEMS [61].

### 2.9. Proteomic Functional Enrichment Analysis

The results from MaxQuant and VEMS searches were processed using Python and then subjected to functional enrichment analysis in R with the clusterProfiler [62] package. Spectral count [63] and intensity-based absolute quantification (iBAQ) [64] values were calculated for all identified proteins. Quantitative data were first preprocessed for normalization. This step was carried out using the ‘normalize.quantiles’ function from the ‘preprocessCore’ R package. Subsequently, the normalized data underwent a log2 transformation, incremented by one, to stabilize the variance and improve the analytical conditions for detecting differential expression. The differential expression analysis itself was conducted utilizing the ‘limma’ R package, setting the ‘trend’ set to FALSE [65]. To select proteins exclusively identified in one experimental group, the selection criteria were based on the spectral count of proteins in both groups, defined as the number of MS/MS spectra associated with peptides from a given protein. For instance, if a protein was associated with at least one spectrum in one group but none in the other group, it was categorized as exclusively identified in the former. Proteins with spectra identified in samples from both groups were considered shared between HFD-PDEV and NCD-PDEV. Genes corresponding to proteins identified with a false discovery rate (FDR) lower than 0.01 underwent over-representation analysis (ORA) [66], using the Gene Ontology [67,68] and KEGG [69,70,71] databases.

### 2.10. Statistical Analysis

Data were analyzed using GraphPad Prism Software, version 9.0.0 (GraphPad Software Inc., San Diego, CA, USA) and presented as mean values with the standard error of the mean (SEM). The significance of the differences between the groups was calculated by unpaired student’s *t*-tests with Welch’s correction. Differences were considered significant as *p*-value ≤ 0.05.

## 3. Results

### 3.1. Evaluation of Diet-Induced Obesity’s Effects on Plasma Derived EVs

To investigate the impact of diet-induced obesity on protein content and PTMs within EVs, we established a murine model designed to replicate obesity hallmarks. A high fat and sugar diet (HFD) was initiated at 6 weeks of age and continued for a duration of 12 weeks to induce obesity in mice (Appendix A). Mice on the HFD exhibited an increase in body weight and hepatic lipid accumulation compared to the control group (Appendix A). To further assess the metabolic impact of obesity, at 11 weeks of diet, both HFD and NCD mice underwent an intra-peritoneal glucose tolerance test (ipGTT). Mice on the HFD displayed reduced glucose removal from the circulation compared to NCD mice, thus indicating impaired glucose tolerance as a consequence of obesity (Appendix A).

Next, we analyzed the presence of proteins, as well as acetylation and glycation events within plasma EVs in the context of obesity. We isolated plasma EVs, which poses a substantial challenge, particularly when dealing with reduced volume samples from murine models. To address this limitation, we pooled samples from 20 animals for each biological replicate. Plasma EVs were characterized by nanoparticle tracking analysis (NTA) and total protein content was quantified by BCA. Both NCD and HFD plasma EVs presented mean sizes ranging between 100 and 150 nm (Appendix A), which falls well within the size range for small EVs, particularly exosomes [15]. We refrained from classifying these EVs as exosomes, since we did not analyze their intracellular origin. NTA and protein quantification revealed a higher particle number and total protein concentration in EVs isolated from HFD mice, resulting in a similar protein content per particle in both experimental groups (Figure 1a–c). Similarly, an increase in circulating EVs under obesity and diabetes was observed in human studies [38,39,40,72].

### 3.2. Proteomic Analysis of Plasma Derived EVs

Next, we analyzed the protein content within plasma EVs isolated from HFD and NCD mice. Trypsinized samples were separated by liquid chromatography, ionized by electrospray ionization, and analyzed by tandem mass spectrometry (Figure 2a). A total of 340 proteins were identified with statistical significance (False Discovery Rate < 0.01). Among these proteins, five were exclusively identified in HFD EVs (Figure 2b,c), 112 proteins were exclusively identified in NCD EVs (Figure 2b, Appendix A); and 223 were common to both groups (Figure 2b). Notably, the vast number of proteins exclusively identified in EVs isolated from NCD-fed animals accounts for a considerable portion of the overall identified proteins. Thus, we observed a decline in the number of protein species carried by plasma EVs in an obesity environment. The 223 proteins identified in both groups were subjected to intensity-based absolute quantification (iBAQ) and differential expression analysis. The resulting log_2_ (fold change) values and their respective −log_10_ (*p*-value) are depicted in Figure 2d. From those, 12 proteins were found to be regulated (*p*-value < 0.05). Within the regulated proteins (Figure 2e), two proteins were upregulated (immunoglobulins), and 10 proteins were downregulated (14-3-3 protein isoforms, proteasome subunits, annexin A7 among others).

Over-representation analysis (ORA) was conducted to gain a better understanding of the identified proteins. In this analysis, we evaluate whether proteins exclusively identified in one experimental group are more enriched in specific functional terms compared to all proteins in that same experimental group, whether exclusively identified or not. This approach allowed us to pinpoint functional terms that are particularly impacted by diet-induced obesity. ORA was conducted using the Gene Ontology database, encompassing three categories: biological processes, cellular components and molecular functions, as well as the KEGG database for both NCD- and HFD-exclusive proteins. However, we observed enrichment of functional terms meeting the significance threshold (adjusted *p*-value < 0.05) solely in the molecular functions ORA for proteins exclusively identified in NCD plasma EVs (Figure 3). This enrichment pointed to an over-representation of functions related to protein folding, phosphatase activity, and hydrolase activity, suggesting the relevance of these functional terms in maintaining metabolic homeostasis. Interestingly, each of these enriched functions was associated with at least 10 out of 16 genes, indicating a high degree of interaction between these functional terms.

### 3.3. Plasma and Gut EVs Crosstalk

Previously, our group revealed changes in the protein content of gut EVs in the context of obesity-associated dysmetabolism [30]. These prior findings hinted at a key role of gut-derived EVs in the spread of the metabolic dysfunction. As such, understanding the intricate crosstalk between plasma and gut EVs becomes paramount to the identification of mechanisms involved in the development of obesity and importantly to the identification of possible biomarkers.

We analyzed the proteins that were shared between plasma and gut EVs (Figure 4a). While no proteins were exclusively detected in the HFD group, we identified four proteins that were exclusive to the control group of both gut and plasma EVs (Figure 4b) (complement factor H, mannose-binding Protein A, Ig heavy chain V region TEPC 1017, and Ig gamma-2A chain C region (secreted form)). The loss of these four proteins in both plasma and gut EVs with the onset of obesity suggests their crucial role in maintaining a healthy homeostatic state and their high potential as biomarkers. Future investigation should address the molecular mechanisms by which their absence may induce or result from obesity.

### 3.4. Post-Translational Modifications (PTMs) in Plasma and Gut EVs Proteins

PTMs act as precise molecular switches capable of modulating proteins’ stability, localization, interactions, and activity. Hence, PTMs enhance the functional complexity of proteins beyond what can be determined solely from their amino acid sequence and folding. To evaluate shifts in PTMs’ profiles, we limited our analysis to modified peptides from proteins shared between HFD and NCD EVs. In the case of plasma EVs, from the 223 proteins shared between both diet groups, nine proteins were acetylated with an FDR < 0.01 (Figure 5a), with four of these proteins being exclusively acetylated in the NCD group (Appendix A). Additionally, we detected 11 glycated proteins with an FDR < 0.01 (Figure 5b), six of which were exclusively glycated in the NCD group (Appendix A). Although the total number of proteins was larger for NCD plasma EVs, these findings suggest a decrease in both acetylation and glycation events in the context of obesity. However, validation of these results is necessary to draw further conclusions.

Regarding gut EVs, a total of 2446 proteins were identified under the significance threshold (FDR < 0.01). Among these proteins, 1804 were shared between HFD and NCD EVs. Next, following the same approach employed for plasma EVs, we focused on acetylated and glycated proteins within the shared set of 1804 proteins. Our analysis revealed 203 acetylated proteins (Figure 5c), with 41 proteins exclusively acetylated in NCD (Appendix A) and 29 proteins exclusively acetylated in HFD (Appendix A). Interestingly, these numbers indicate a decrease in acetylation events under obesity, mirroring the observations made for acetylated proteins in plasma EVs. For glycation, a total of 56 proteins were found to be glycated in gut EVs (Figure 5d), with 23 proteins exclusively glycated in NCD (Appendix A) and 22 proteins exclusively glycated in HFD (Appendix A).

Acetylated and glycated proteins exclusively modified in either NCD or HFD gut derived EVs were subjected to ORA. For these analyses, the list of 1804 proteins identified in both NCD and HFD gut EVs was used as the background reference. Considering Gene Ontology for cellular compartments (Figure 6a), proteins exclusively acetylated in HFD gut EVs and proteins exclusively glycated in NCD gut EVs exhibited significant enrichment in protein-DNA complexes, nucleosome, chromatin, and polymerase complexes, while proteins exclusively glycated in HFD gut EVs were primarily associated with the cytoskeleton and actin filaments. Regarding Gene Ontology for molecular functions (Figure 6b), proteins exclusively acetylated in HFD gut EVs were enriched in structural constituents of the chromatin, while proteins exclusively glycated in NCD gut EVs exhibited a significant enrichment in structural constituents of the chromatin, DNA binding and protein heterodimerization. In addition, proteins exclusively glycated in HFD gut EVs showed significant enrichment in structural constituents related to the cytoskeleton, synapse, and post synapse.

Considering KEGG pathways (Figure 6c), proteins exclusively acetylated in HFD gut EVs and proteins exclusively glycated in NCD gut EVs exhibited significant enrichment in pathways related to viral carcinogenesis, systematic lupus erythematosus, alcoholism, and neutrophil extracellular trap formation. It is noteworthy that histones were associated with all four of these enriched KEGG pathways (Appendix A). On the other hand, proteins exclusively glycated in HFD gut EVs were enriched in terms that were mainly related to apoptosis, tight junctions, motor proteins, and the phagosome. Importantly, in gut EVs, proteins exclusively glycated in HFD, the most enriched functional term across these three analyses were consistently linked to actin and tubulin (Appendix A).

Altogether, these findings indicate that EVs transport proteins with several PTMs and this profile is altered by obesity. Our proteomic data can serve as the basis to develop future studies to understand how PTMs of proteins present in EVs can be modulated to contain obesity progression.

## 4. Discussion

Given the rapidly changing global dietary patterns, the prevalence of obesity and its associated health conditions continues to rise. Implementing interventions to combat obesity could serve as a potent strategy to mitigate multiple diseases that exert a significant socioeconomic impact. Hence, it is crucial to understand the molecular mechanisms underlying obesity for developing efficacious treatments. As obesity involves systemic interactions depending on inter-organ communication, EVs play a crucial role. Importantly, several studies have highlighted the relevance of EV cargo in maintaining metabolic homeostasis and enabling cell communication [16,18,19,20]. The protein content of EVs varies depending on the organ they originate from. As a result, EVs found in the bloodstream comprise a diverse mix originated from different organs, aiding in inter-organ communication. The possibility of isolating EVs from body fluids enhances their potential as biomarkers for diagnosis and/or prediction of treatment outcomes. Moreover, they serve as therapeutic tools for various diseases. In this study, we present the first proteomics analysis of plasma EVs isolated from obese animals and their overlap with gut EVs. 

The obese male murine model we used not only exhibits increased body weight, but it also reflects obesity comorbidities, such as the hallmark features of prediabetes exemplified by glucose intolerance and liver steatosis indicated by hepatic lipid deposition (Appendix A). Although studies of gender impact on EV content are scarce and report inconsistent information, some have reported gender-related differences in protein, lipid, and RNA composition, and in physiological and pathological conditions. Therefore, the exclusive use of male mice limits the generalizability of our findings [73,74,75,76,77]. Consistent with findings from human studies [72], we observed an increase in the number of plasma EVs under obesity conditions (Figure 1). A total of 340 proteins were identified, with five being exclusive to HFD plasma EVs and 112 exclusive to NCD EVs (Figure 2). Although EV samples were isolated through a rigorous protocol that includes multiple rounds of ultracentrifugation and a sucrose cushion, complete elimination of contaminant proteins remains a challenge. We detected the presence of ApoA2 and immunoglobulins, both of which are present in high levels in plasma [78,79,80]. Nevertheless, our findings suggest a reduction in cargo diversity in diet-induced obesity, indicating that the messaging system becomes more refined and intense, with an increased number of circulating EVs, or that cells try to compensate the lack of protein diversity with heightened EVs secretion. Either way, this may represent an adaptive mechanism or a response to ongoing metabolic changes. The key finding here is that plasma EVs encapsulate proteins that are lost as the disease progresses, highlighting their crucial role in maintaining metabolic homeostasis. These proteins could provide insights into the mechanisms underlying obesity. 

A further analysis using ORA unveiled a significant enrichment of proteins associated with protein-folding processes among the 112 proteins exclusively identified in plasma EVs isolated from control animals, particularly chaperonin-containing TCP1 and heat-shock proteins (Figure 3). Indeed, the functionality of a protein is intrinsically linked to its conformation, and anomalies in protein folding lead to a range of alterations associated with the etiology of various human diseases, including β-cell dysfunction [81,82,83,84]. Important examples of this association include the islet amyloid polypeptide, which forms toxic aggregates leading to β-cell dysfunction and death [82,83], and heat shock factor 1, a transcriptional activator of molecular chaperones involved in the development of T2D [84]. 

Regarding the 12 regulated proteins identified in plasma EVs common to both NCD and HFD (Figure 2), it is worth noting that the 14-3-3 proteins presented higher expression in NCD plasma EVs, whereas immunoglobulins presented higher expression in HFD plasma EVs (Figure 2d,e). The 14-3-3 proteins are molecular adaptors which regulate a broad spectrum of signaling pathways and, more importantly, are known to have a beneficial effect on β-cell function and survival, which may be comprised in prediabetes [85]. On the other hand, the increased expression of immunoglobulins in HFD plasma EVs could indicate the onset of subclinical chronic inflammation, characteristic of obesity [86].

One of the novel aspects of this study is the analysis of protein PTMs within EVs, aiming to identify signaling molecules that are acetylated and glycated. These molecules have the potential to directly influence signaling functions within the recipient cells, either enhancing or inhibiting them. PTMs act as a dynamic and finely tuned molecular switches that can significantly alter protein function, stability, and interactions [87,88]. Our investigation into acetylated and glycated proteins within plasma EVs revealed a trend—obesity was associated with decreased acetylation and glycation events (Figure 5a,b). This observation suggests that the regulation of these PTMs is altered by the onset of obesity. The similarity between proteins exclusively acetylated in gut EVs in the HFD group and proteins exclusively glycated in the NCD group suggests that these proteins, mainly histones, undergo regulatory shifts by PTMs under obesity conditions. Importantly, this regulation of PTMs is detectable within the intercellular communication network, indicating their relevance in the context of organ crosstalk in obesity. Given the well-established roles of PTMs in insulin signaling, glucose metabolism, and the development of associated complications [42,43,44], this finding opens up new avenues for understanding the molecular mechanisms underlying obesity. Importantly, it calls our attention to the need to not only to examine the presence of a given protein but also consider its PTM status. Further research into the functional consequences of these PTM alterations in the context of metabolic health and disease progression is warranted.

Expanding on our previous study of the protein cargo within gut EVs in obesity, we reanalyzed the data focusing on PTMs. We observed a decrease in acetylation events associated with obesity, indicating a potentially intricate regulatory mechanism for these PTMs that may involve interactions with other organs (Figure 5c). When analyzed by ORA, the 29 proteins exclusively acetylated in HFD gut EVs showed an enrichment in GO functional terms related to chromatin, nucleosome, and DNA transcription, with histones being the main enriched proteins (Figure 6a,b and Appendix A). These proteins interact with DNA and provide structural support to chromatin, regulating its condensation according to their own PTM profile [89]. While these associations suggest a crucial role for acetylation, especially of histones, thus controlling transcription in regulating obesity, further research is needed to unravel the precise mechanisms and functional implications of these observations in the context of metabolic health and disease. More intriguingly, the enriched functional terms obtained from KEGG pathways include viral carcinogenesis, systemic lupus erythematosus, alcoholism, and neutrophil extracellular trap formation (Figure 6c). These four pathways have been associated with histone acetylation, suggesting their possible interconnection with the development of obesity [90,91,92,93,94].

After analyzing the 23 proteins exclusively glycated in NCD gut EVs using ORA, the results for GO and KEGG were very similar to those for proteins exclusively acetylated in HFD EVs (Figure 6), even with some proteins shared between the two conditions (Appendix A). This not only indicates that these enriched functions are impacted by obesity, but also strongly suggests that they are regulated by a shift in histones’ PTM profile. More precisely, these changes could involve an interplay between two types of modifications: glycation and acetylation. This interplay suggests that histone acetylation and glycation may compete for the same sites, since both modifications primarily occur on lysine and arginine residues [95,96,97]. Exploring the specific glycation and acetylation sites of these proteins will be crucial to pinpoint the regulatory mechanism of inter-organ communication mediated by EVs under prediabetic conditions.

ORA enrichment of the 22 proteins exclusively glycated in HFD gut EVs revealed several GO terms related to cellular structural stability (Figure 6). The most enriched proteins were actin and tubulin (Appendix A). Glycation of both proteins leads to an increase in endothelial permeability, which compromises the gut barrier during T2D [98,99,100]. ORA enrichment also revealed KEGG pathways associated with cellular structure and integrity. The enriched pathways of amyotrophic lateral sclerosis, dilated cardiomyopathy and hypertrophic cardiomyopathy are potentiated by increases in blood central nervous system barrier permeability and sarcolemmal permeability, respectively [101,102,103]. It is possible that gut EVs are transporting glycated structural proteins and delivering them to other organs, thereby increasing the risk of developing such pathologies. Further research is needed to explore the mechanistic details and functional implications of these observations.

The crosstalk between plasma and gut EVs adds an extra layer of complexity to our understanding of obesity. In this study, we identified four proteins exclusively present in both plasma and gut EVs under healthy conditions (Figure 4a,b). Their absence in obesity raises questions about their role in maintaining metabolic homeostasis. These proteins possess high biomarker potential for a healthy physiological state, as their presence is lost in obesity. Notably, the term “immune response” emerged as significant when examining the function of individual proteins, suggesting that disruption in immune system regulation may play a crucial role in metabolic dysfunction [104]. Complement factor H, one of these proteins, has previously been linked to metabolic disorders, reinforcing its relevance in the context of prediabetes [105]. The detection of proteins shared between plasma and gut EVs highlights the possibility of a non-invasive method to evaluate gut health through the analysis of blood-derived EVs. However, most proteins identified in gut EVs were not found in plasma EVs. We attribute this primarily to anatomical and physiological factors. Gut EVs are released into the portal vein, which directly drains into the liver, where they may be up taken by liver cells, reducing their contribution for the overall population of EVs in systemic circulation. Moreover, the exact mechanisms by which these altered protein cargos and PTMs contribute to obesity progression remain elusive. Further research is needed to elucidate these mechanisms, potentially through functional studies.

## 5. Conclusions

This study illuminates the novel role of EVs in obesity and their potential as biomarkers for early metabolic dysfunction. The changes in protein composition and PTMs offer new perspectives into the molecular mechanisms behind obesity. These findings have the potential to pave the way for early intervention strategies to combat the rising prevalence of obesity, emphasizing the critical role of EVs in understanding and addressing these health challenge. Further research in this growing field promises to unravel more intricacies and insights, ultimately advancing our ability to prevent and treat obesity more effectively.

## Figures and Tables

**Figure 1 nutrients-16-00736-f001:**
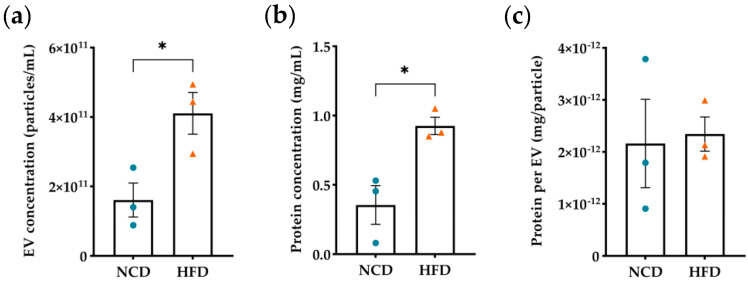
Diet-induced obesity increases the number of plasma extracellular vesicles. (**a**) Analysis of the number of particles per mL of sample from control (NCD) and diet-induced obese mice (HFD). (**b**) Protein quantification in plasma EVs in mg/mL, obtained by BCA. (**c**) Protein content per EV, represented in mg of protein per particle. All statistical analysis were performed using unpaired *t*-test with Welch’s correction. All data are presented as mean ± standard error of the mean. * *p*-value < 0.05. Each *n* represents a sample of 20 animals, *n* = 3 for both groups, with blue dots representing NCD samples and orange triangles representing HFD samples.

**Figure 2 nutrients-16-00736-f002:**
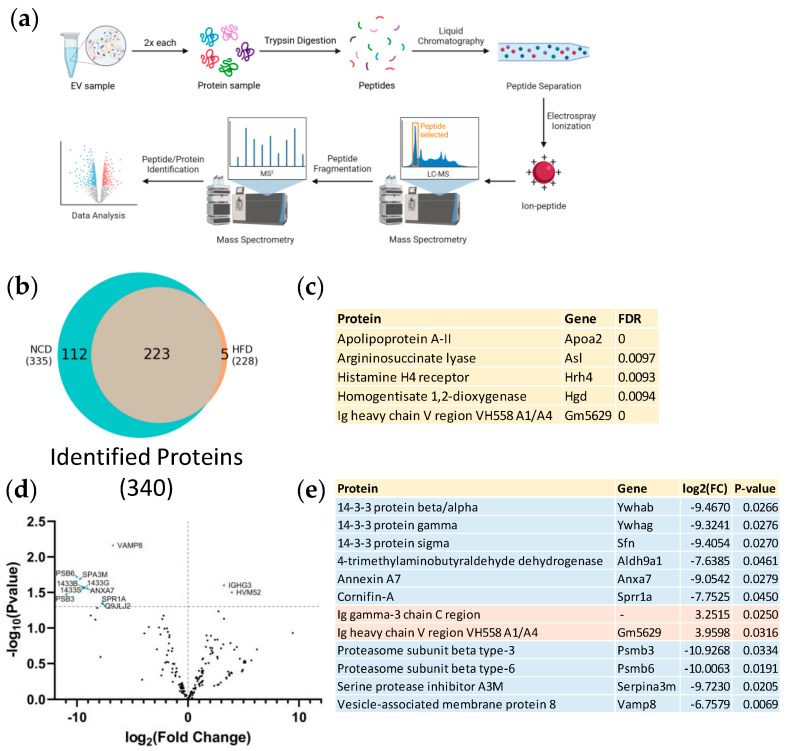
Comparative protein profile in plasma-derived extracellular vesicles from NCD and HFD-fed mice. (**a**) Schematic representation illustrating the experimental workflow for the analysis of plasma EVs. Each sample, obtained from mice fed either an NCD or an HFD, underwent two rounds of analysis using liquid chromatography-tandem mass spectrometry (LC-MS/MS). (**b**) Venn diagram representing the overlap of proteins identified in NCD and HFD plasma EVs with a false discovery rate (FDR) < 0.01. (**c**) List of proteins exclusively identified in HFD plasma EVs. (**d**) Volcano plot representing the fold change and *p*-value of regulated proteins shared between NCD and HFD plasma EVs. Dotted horizontal line indicates the threshold for a *p*-value < 0.05. A positive log_2_ (fold change) value indicates higher protein levels in diet-induced obese mice when compared to control mice. Down-regulated proteins are represented in light blue, while up-regulated proteins are indicated in light red. (**e**) List of significantly regulated proteins (*p*-value < 0.05) in plasma EVs from HFD mice compared to NCD mice.

**Figure 3 nutrients-16-00736-f003:**
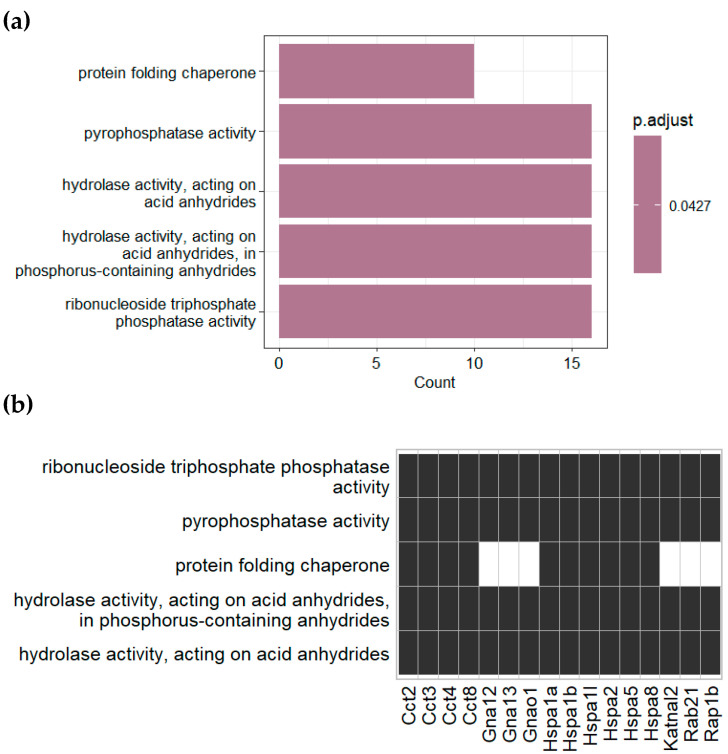
Gene Ontology over-representation analysis of plasma EVs proteins identified exclusively in NCD mice. (**a**) Bar plot and (**b**) table of enriched molecular functions and their related identified genes. Black rectangles indicate the association between the identified gene and the enriched term.

**Figure 4 nutrients-16-00736-f004:**
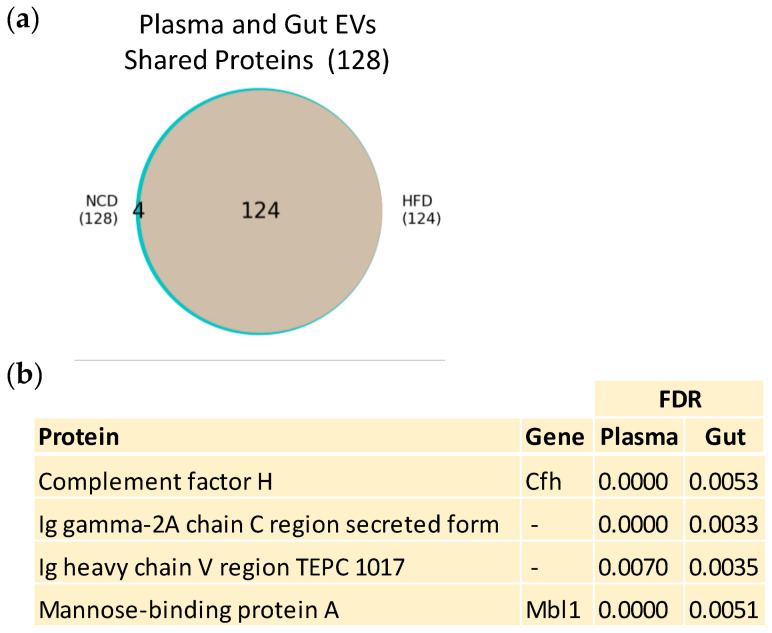
Obesity-associated changes in proteins shared between plasma and gut EVs. (**a**) Venn diagram displaying the intersection of proteins identified in HFD mice and NCD mice, from all proteins shared between PDEV and GDEV. All proteins presented an FDR < 0.01. (**b**) List of four proteins shared between PDEV and GDEV, present exclusively in NCD mice.

**Figure 5 nutrients-16-00736-f005:**
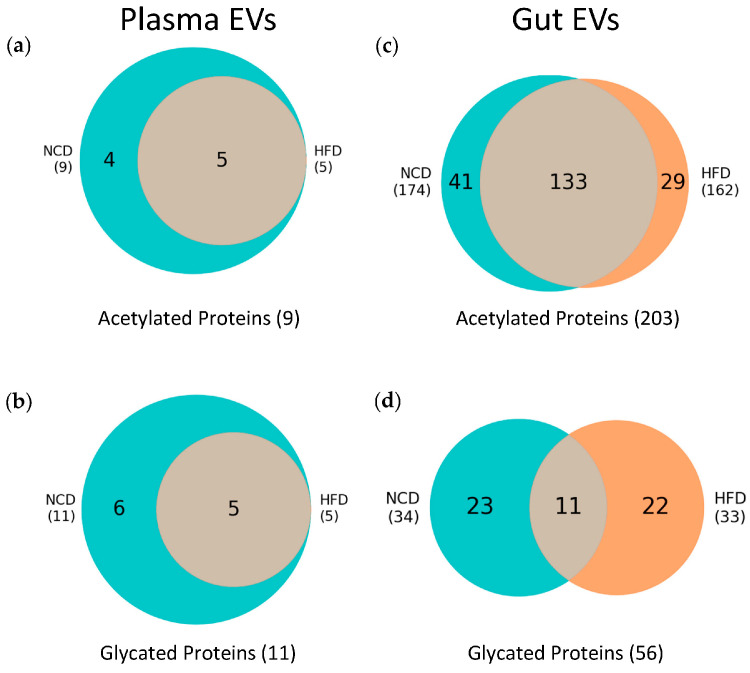
Analysis of protein acetylation and glycation in plasma and gut EVs. (**a**) Venn diagram representing the intersection of acetylated proteins identified in NCD and HFD plasma EVs. (**b**) Venn diagram representing the intersection of glycated proteins identified in NCD and HFD plasma EVs. (**c**) Venn diagram representing the intersection of acetylated proteins identified in NCD and HFD gut EVs. (**d**) Venn diagram representing the intersection of glycated proteins identified in NCD and HFD gut EVs. All proteins presented an FDR < 0.01. The identification of acetylated and glycated proteins was only conducted on proteins previously identified in both NCD and HFD samples (223 proteins for plasma EVs and 1804 proteins for gut EVs).

**Figure 6 nutrients-16-00736-f006:**
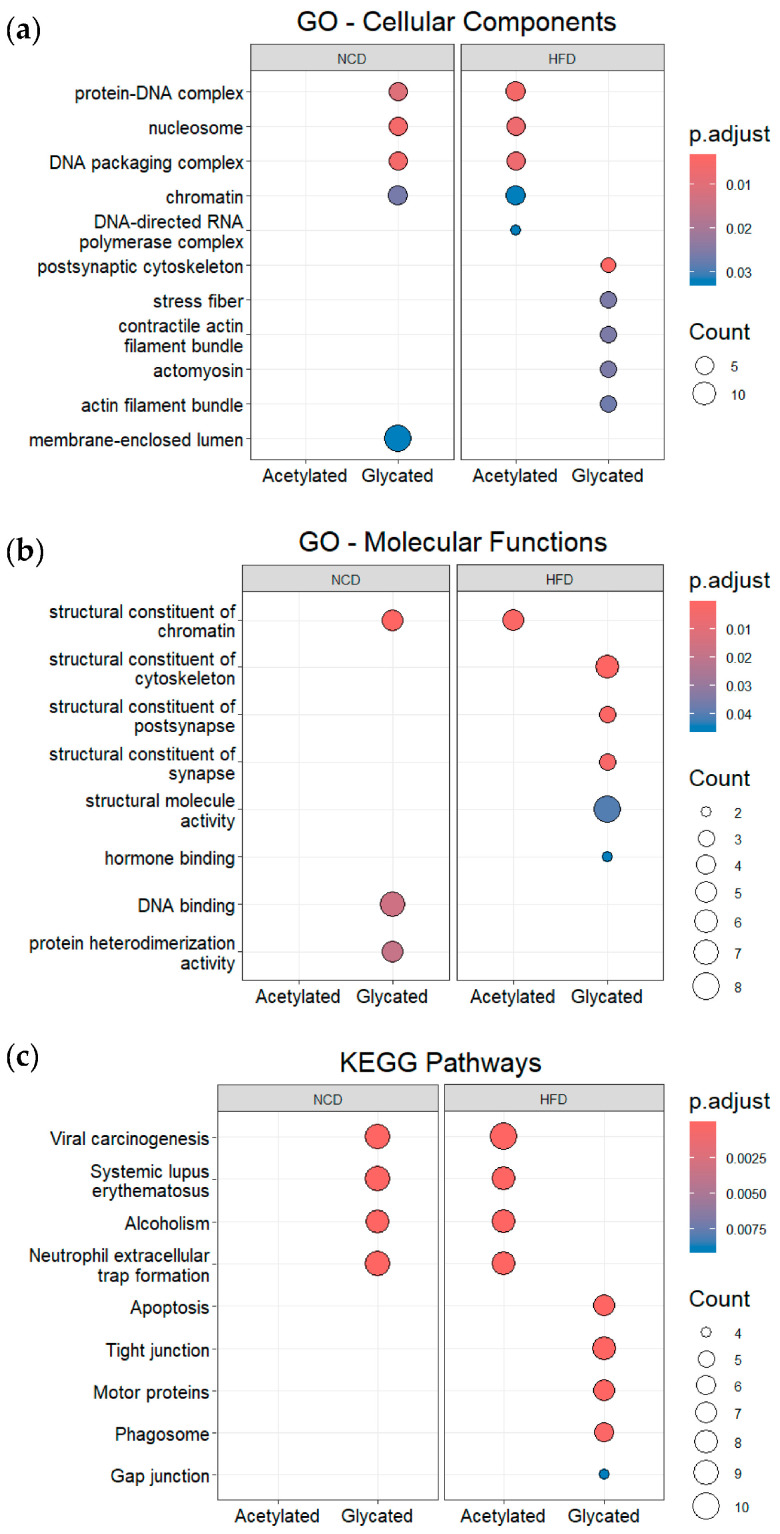
Over-representation analysis of gut EVs proteins featuring acetylation and glycation. (**a**) GO enrichment analysis according to cellular components. (**b**) GO enrichment analysis according to molecular functions. (**c**) KEGG enrichment analysis for KEGG pathways. Dot plots represent the five most significantly enriched terms in each group.

## Data Availability

Data are contained within the article and Appendix A.

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
