# Peer review of "Proteomic Profiling of Plasma- and Gut-Derived Extracellular Vesicles in Obesity"

_nutrients, 2024, doi:10.3390/nu16050736_

Round 1

Reviewer 1 Report

Comments and Suggestions for Authors

Pereira et al. examined the proteomic profiling of plasma EVs from diet-induced obese mice, compared the plasma EV data with published gut EV data, and identified four EV proteins as putative biomarkers.  It’s an interesting study that can provide some information to the field, however, additional functional study is needed, and the experimental design and the interpretation of data should be clarified and improved.

Concerns:

1. Functional analysis of at least one of the four candidate proteins should be performed in vivo. For instance, determine the therapeutic effects of i.p. injection of candidate protein into diet-induced obese mice.

2. It’s unclear the rationale for comparing plasma EVs with gut EVs.  Why not compare with EVs derived from other tissues?

3. The reason why the majority of gut EVs were not detected in the plasma needs to be discussed.  What are the origins of the plasma EVs?  Why gut EVs do not contribute to plasma EVs?

4. Diet-induced obese mouse model has been widely reported, so the data of model characterization should be moved to supplementary.

Comments on the Quality of English Language

Scientific wiring needs to be improved.  The rationale for each experiment should be shown clearly.

Author Response

Thank you very much for taking the time to review our manuscript and for providing constructive and valuable feedback aimed at improving the quality of our publication. Please find the detailed responses below. Additionally, the corresponding revisions and corrections have been highlighted or indicated via track changes in the re-submitted files for your convenience.

  1. Functional analysis of at least one of the four candidate proteins should be performed in vivo. For instance, determine the therapeutic effects of i.p. injection of candidate protein into diet-induced obese mice.

We greatly appreciate the reviewer’s insightful suggestion and recognize the potential value of conducting in vivo functional analysis of the candidate proteins. However, we believe such studies fall outside the scope of the current publication. As detailed throughout our manuscript, given the pivotal role of EVs in facilitating intercellular and inter-tissue communication through their molecular cargo, the primary goal of the current research is to employ proteomics to uncover novel pathways associated with obesity that are communicated by EVs. This approach opens avenues for future research, setting a foundation for subsequent functional analyses, as proposed by the reviewer. Furthermore, it is important to highlight the innovative aspect of our study. Previous research on the EV content associated with diabetes and obesity predominantly focused on microRNAs (miRNAs) within adipose tissue. In contrast, our work extends beyond proteins to examine changes in their post-translational modifications (PTMs) and explores EVs content from both the gut and plasma, underscoring the significance of the gut in this context.

Regarding the suggestion to follow up with in vivo testing of candidate proteins via intraperitoneal injection, we plan to consider elements of this recommendation in future publications. However, we have reservations about the direct therapeutic applicability of i.p. injections of the proteins not protected inside the EVs. Proteins within EVs are shielded by the vesicle's double membrane, thus protecting them from blood-borne proteases. Moreover, for therapeutic efficacy, these proteins need targeted delivery to specific organs, a process facilitated by the unique surface repertoire of EVs. Interestingly, a recent study by Miotto et al. (Nat Metab, 2024, https://doi.org/10.1038/s42255-023-00971-z) demonstrated that mice injected with sonicated EVs, which breaks the EVs’ double membrane and releases their content, did not exhibit the same therapeutic effects as those injected with intact EVs. Remarkably, the outcomes from sonicated EVs mirrored those of the saline control, reinforcing the importance of intact EVs for desired therapeutic outcomes.

In summary, while we acknowledge the value of the proposed functional analysis, our current focus on identifying and characterizing novel protein pathways in obesity via EVs lays the groundwork for such future investigations.

  1. It’s unclear the rationale for comparing plasma EVs with gut EVs.  Why not compare with EVs derived from other tissues?

Thank you for bringing attention to the need for further clarifying the rationale behind comparing these two sources of EVs. Our primary focus was on analyzing plasma EVs because (1) they represent EVs released from various tissues and (2) of their non-invasive collection, which allows for liquid biopsies to be performed. The choice to focus on plasma and gut EVs aligns with our goal to explore the systemic impact of diet-induced obesity, which is less emphasized in the existing literature predominantly centered on white adipose tissue. In comparing plasma EVs to gut EVs, we aim to explore the gut's pivotal role in metabolic regulation and obesity. The gut is a critical mediator in maintaining metabolic homeostasis, influenced significantly by dietary intake, exposure to pathogens, and interaction with the host's immune system. Our decision is further supported by the gut's proven involvement in effective obesity interventions, such as bariatric surgery, and the emerging interest in gut microbiota's role in obesity management through approaches like fecal microbiota transplantation.

This comparative analysis is not only a continuation but an expansion of our lab group's previous research (doi.org/10.1021/acs.jproteome.1c00353), which highlights the gut's crucial contribution to the conversation on obesity. By focusing on the gut, we aim to integrate EVs into the broader dialogue on gut-mediated communication mechanisms, which classically only included the nervous system and hormones.

In toto, the rationale for our comparative approach between plasma and gut EVs, including the gut's central role in obesity and metabolic regulation, is detailed in the introduction of our manuscript, particularly between lines 72 to 83.

  1. The reason why the majority of gut EVs were not detected in the plasma needs to be discussed.  What are the origins of the plasma EVs?  Why gut EVs do not contribute to plasma EVs? 

We thank the reviewer for pointing out the discrepancy in the detection of gut EVs proteins in plasma EVs. To address this, we have edited the manuscript to include a discussion on this topic (lines 558-563). As outlined in our response to a previous question, plasma contains different sub-populations of EVs that originate from nearly every organ and cell type. However, the contribution of each organ to the total amount of circulating EVs is an open question in the field. We believe the disparity observed stems primarily from a physiological and anatomical reason. EVs released from the gut enter the portal circulation, directing them first to the liver. This passage suggests a potential filtration or sequestration mechanism within the liver, which could significantly limit the number of gut EVs reaching the systemic circulation. Supporting this hypothesis, our unpublished results have identified Kupffer cells in the liver as the primary cells capturing gut EVs. The fact that gut EVs are directly captured by the liver macrophages upon their exit from the gut, likely explains the low availability and detection of gut EVs in the systemic circulation. 

  1. Diet-induced obese mouse model has been widely reported, so the data of model characterization should be moved to supplementary. 

We acknowledge and agree with this suggestion and we moved model characterization data from Figure 1 into a new supplementary figure (Figure S3).

  1. The rationale for each experiment should be shown clearly. 

We acknowledge and agree the reviewer's emphasis on the importance of clearly presenting the rationale behind each experiment. In response, we have carefully reviewed our manuscript and ensured that the rationale for each key experiment is explicitly stated and easily identifiable for the reader. Thus, we summarize below, what is found in results section, and we believe are the rationale behind each step, clearly stated.

3.1. Evaluation of diet-induced obesity’s effects on plasma derived EVs

To investigate the impact of diet-induced obesity on protein content and PTMs within EVs, we established a murine model designed to replicate obesity hallmarks. (Lines 235-236).

Next, we analyzed the presence of proteins, as well as acetylation and glycation events within plasma EVs in the context of obesity (Lines 245-246).

3.2. Proteomic analysis of plasma derived EVs

Next, we analyzed the protein content within plasma EVs isolated from HFD and NCD mice. (Lines 285-286).

Over-representation analysis (ORA) was conducted to gain a better understanding of the identified proteins. In this analysis we evaluate if proteins exclusively identified in one experimental group are more enriched in specific functional terms compared to all proteins in that same experimental group, whether exclusively identified or not. This approach allows to pinpoint functional terms that are particularly impacted by diet-induced obesity (Lines 315 – 320).

3.3. Plasma and Gut EVs crosstalk

Previously our group revealed changes in the protein content of gut EVs in the context of obesity dysmetabolism [31]. These prior findings hinted to a key role of gut derived EVs in the spread of the metabolic dysfunction. As such, understanding the intricate crosstalk between plasma and gut EVs becomes paramount to the identification of mechanisms involved in the development of obesity and importantly to the identification of possible biomarkers. (Lines 336 – 341).

3.4. Post-translational modifications (PTMs) in plasma and gut EVs proteins

PTMs act as precise molecular switches capable of modulating protein's stability, localization, interactions and activity. Hence, PTMs enhance proteins’ functional complexity beyond what can be determined solely from their amino acid sequence and folding. To evaluate shifts in PTMs’ profiles, we limited our analysis to modified peptides from proteins shared between HFD and NCD EVs (Lines 357 -361).

Reviewer 2 Report

Comments and Suggestions for Authors

In this manuscript, Pereira et al. carried out a proteomic analysis of the protein cargo in extracellular vesicles (EVs) obtained from plasma and gut of HFD and chow fed mice. They identified unique and differentially expressed proteins and also performed an analysis of the changes in PTM as a function of obesity. Overall, this is a straightforward paper with some useful and interesting data on proteomic changes associated with obesity. However, there are some concerns with the observed results and the potential of confounding. To address these concerns, the authors are requested to consider the following points:

1. Lines 259-274 are better placed under Discussion, rather than under Results. Please focus the Results section to only describe and interpret the results.

2. The studies are done in male mice only. It will be useful to know how HFD affects EVs in female mice. If the authors can generate data that will be great. Otherwise, they may refer to other publications on EVs in female mice if they exist. Else, they should point this out as a limitation on the generalizability of their results.

3. I am unclear what the authors mean by protein ‘diversity’ on line 288. If by diversity they mean different classes of proteins, then it is perhaps a bit hasty to conclude that just because there were less unique proteins in HFD compared to NCD EVs, the diversity of proteins were also less under HFD. The proteins could be just as diverse in the HFD group but with lower number of protein representatives for each diverse group. One way to test this is to use something like the Panther webtool (www.pantherdb.org) to separate the proteins from NCD and HFD EVs into categories and compare their diversity that way.

4. The authors reported the FDR when analyzing proteins that are expressed (line 283) but report nominal p-values when analyzing for differential expression (line 292). Please report FDR for differential protein expression analysis also.

5. It appears that limma was used for differential protein analysis (line 210). Please furnish more details on the parameters used in limma. For example, was the data pre-normalized before applying limma? Was limma-trend or limma-voom used for differential expression analysis?

6. Figure legend 2d has reference to ‘prediabetic’ mice. This is most certainly an error that needs to be corrected.

7. Line 216 - instead of saying ‘Genes of proteins identified…’ it might be better to say ‘Genes corresponding to proteins identified…’.

8. Separation of EVs from highly abundant free plasma proteins is a concern in these types of proteomics studies. For example, ApoA2 and Ig heavy chain V are also present in high levels in the plasma and thus their ‘unique’ presence in HFD EVs could be actually a confounded finding (Figure 2c). The authors should conduct some experiments to disentangle this (e.g. see doi.org/10.7554/eLife.86394) or include this caveat as a limitation of the study in the Discussion section.

9. I wasn’t sure if the results reported for the gut EVs were new findings and not a repetition from the authors’ earlier work (reference 31). Kindly clarify.

10. The value of the PTM-based analysis remains unclear to me. It is not just the PTM, but rather the site(s) of the PTM on the protein that has functional relevance. Also, since PTM is a dynamic event, we probably have only limited information on the full scale of PTM observed for any given protein. Additionally, it is also unknown if the PTM modifications observed in EVs is a true reflection of the PTMs present on the functionally active protein (both qualitatively and quantitatively). Overall, this makes me less enthusiastic to consider the PTM-related results at their current form.

Comments on the Quality of English Language

Minor editing needed.

Author Response

Thank you very much for taking the time to review our manuscript and for providing constructive and valuable feedback aimed at improving the quality of our publication. Please find the detailed responses below. Additionally, the corresponding revisions and corrections have been highlighted or indicated via track changes in the re-submitted files for your convenience.

Lines 259-274 are better placed under Discussion, rather than under Results. Please focus the Results section to only describe and interpret the results.

We appreciate your suggestion regarding the placement of lines 259-274. Upon review, we agree that these sentences are more appropriately situated within the Discussion section. Accordingly, we have moved these sentences from the Results to the Discussion section. We believe this modification enhances the overall clarity and logical flow of our manuscript. Thank you for guiding this improvement.

The studies are done in male mice only. It will be useful to know how HFD affects EVs in female mice. If the authors can generate data that will be great. Otherwise, they may refer to other publications on EVs in female mice if they exist. Else, they should point this out as a limitation on the generalizability of their results. 

We greatly appreciate the reviewer’s insightful comment regarding the inclusion of only male mice in our studies and the potential impact of a high-fat diet on EVs in female mice. We fully recognize the importance of gender, unfortunately the current knowledge is not much developed. Indeed, existing literature suggests that there may be gender-specific variations in the proteome of EVs, highlighting the importance of this consideration. Incorporating female mice would imply establishing new animal cohorts, which takes around 5 months, and all the subsequent work, which is beyond our current capabilities. To address this limitation, we have included a sentence in the discussion section (438-445) that acknowledges the study's focus on male mice and the potential implications for generalization.

I am unclear what the authors mean by protein ‘diversity’ on line 288. If by diversity they mean different classes of proteins, then it is perhaps a bit hasty to conclude that just because there were less unique proteins in HFD compared to NCD EVs, the diversity of proteins were also less under HFD. The proteins could be just as diverse in the HFD group but with lower number of protein representatives for each diverse group. One way to test this is to use something like the Panther webtool (www.pantherdb.org) to separate the proteins from NCD and HFD EVs into categories and compare their diversity that way.

We appreciate the reviewer's insightful suggestion and the opportunity to clarify what we meant by 'protein diversity' in our manuscript. We recognize that our initial use of the term 'diversity' may have unintentionally suggested a broader analysis of protein classes than was conducted. To rectify this, we have revised our manuscript accordingly. Specifically, we have replaced the term 'diversity' with 'number of protein species' to more accurately reflect that our analysis was focused on the count of distinct protein species identified in each group (NCD vs. HFD), rather than an assessment of the variety of protein classes or groups.

The authors reported the FDR when analyzing proteins that are expressed (line 283) but report nominal p-values when analyzing for differential expression (line 292). Please report FDR for differential protein expression analysis also.

We appreciate the reviewer's attention to the statistical methods applied in our analysis and the request for reporting the false discovery rate (FDR) for differential protein expression. In our analysis, we indeed focused on nominal p-values. We acknowledge that after applying FDR correction for multiple testing, no proteins were identified as significantly regulated. This observation was anticipated due to the relatively small sample size, which may limit the statistical power necessary to detect significant differences after correcting for multiple comparisons.

While we understand the importance of FDR, we chose to proceed with the functional analysis based on proteins identified through nominal p-values. This decision was made under the consideration that the insights gained could provide valuable overall pathway regulation and guide future research directions with the assumption that a few false positive regulated proteins should not dramatically affect the overall functional regulation.

We hope this explanation clarifies our approach and the rationale behind. We believe that despite the limitations, our study contributes valuable insights into the proteomic changes associated with obesity and lays the groundwork for future investigations with larger cohorts that could overcome these constraints.

It appears that limma was used for differential protein analysis (line 210). Please furnish more details on the parameters used in limma. For example, was the data pre-normalized before applying limma? Was limma-trend or limma-voom used for differential expression analysis?

We appreciate the reviewer's comment and the opportunity to clarify the analytical methods used in our study. To address your inquiry regarding the use of the 'limma' package for differential protein analysis, we have expanded the description in the Methods section of our manuscript. The revised text now specifies the preprocessing steps and the parameters employed in our 'limma' analysis as follows: "Quantitative data were first preprocessed for normalization. This step was carried out using the 'normalize.quantiles' function from the 'preprocessCore' R package. Subsequently, the normalized data underwent a log2 transformation, incremented by one, to stabilize the variance and improve the analytical conditions for detecting differential expression. The differential expression analysis itself was conducted utilizing the 'limma' R package, setting the 'trend' set to FALSE."

Figure legend 2d has reference to ‘prediabetic’ mice. This is most certainly an error that needs to be corrected. 

Thank you for highlighting this mistake regarding the incorrect reference to 'prediabetic' mice in Figure 2d legend. The term 'prediabetic' has been corrected to 'diet-induced obese' to accurately reflect the condition of the mice used in our experiments. We appreciate your attention to detail and assistance in improving the accuracy of our manuscript

Line 216 - instead of saying ‘Genes of proteins identified…’ it might be better to say ‘Genes corresponding to proteins identified…

We appreciate the reviewer's suggestion to refine the phrasing for greater clarity. The phrase has been updated to 'Genes corresponding to proteins identified...' as suggested. One more time, we would like to thank you for helping us enhance the quality of our work.

Separation of EVs from highly abundant free plasma proteins is a concern in these types of proteomics studies. For example, ApoA2 and Ig heavy chain V are also present in high levels in the plasma and thus their ‘unique’ presence in HFD EVs could be actually a confounded finding (Figure 2c). The authors should conduct some experiments to disentangle this (e.g. see doi.org/10.7554/eLife.86394) or include this caveat as a limitation of the study in the Discussion section. 

We appreciate the reviewer's concern regarding the separation of EVs from highly abundant free plasma proteins, a challenge in EVs proteomics. It is indeed crucial to differentiate between proteins present in EVs and those that may be co-isolated due to their abundance in plasma, such as ApoA2 and Ig heavy chains. To address the specific concern raised, we compared our methodology against the one suggested in the referenced paper (doi.org/10.7554/eLife.86394). While the paper suggests a size exclusion chromatography-based method, we are using a protocol which involves a meticulous series of ultracentrifugation steps. Noteworthy, both protocols incorporate a sucrose density cushion, as an additional step believed to enhance the purity of EVs preparations and reduce the co-isolation of free plasma proteins. However, we acknowledge that no method can completely eliminate the presence of contaminant proteins in EVs preparations, and this limitation applies universally across the field.

To transparently address the potential for confounding findings due to the presence of highly abundant plasma proteins in our EVs samples, we have discussed this as a limitation of our study in lines 448-452.

I wasn’t sure if the results reported for the gut EVs were new findings and not a repetition from the authors’ earlier work (reference 31). Kindly clarify.

We appreciate the opportunity to clarify this aspect of our study. The results presented for gut EVs indeed constitute new findings and represent a significant extension of our earlier work cited as reference 31. While our previous publication laid the groundwork by characterizing gut EVs in the context of diet-induced obesity, the current study advances our understanding in two critical ways.

1) In the present study, we intersect our previously published data on gut EVs with our comprehensive analysis of plasma EVs obtained from the same dietary intervention model.

2) The current manuscript introduces a novel study of post-translational modifications (PTMs) within proteins carried by both gut and plasma EVs. This analysis of PTMs represents a novel dimension of the current research. The identification and characterization of these PTMs are entirely new contributions to the field and were not addressed in our previous work.

In summary, this was a novel approach on our part and all resulting findings are entirely novel.

The value of the PTM-based analysis remains unclear to me. It is not just the PTM, but rather the site(s) of the PTM on the protein that has functional relevance. Also, since PTM is a dynamic event, we probably have only limited information on the full scale of PTM observed for any given protein. Additionally, it is also unknown if the PTM modifications observed in EVs is a true reflection of the PTMs present on the functionally active protein (both qualitatively and quantitatively). Overall, this makes me less enthusiastic to consider the PTM-related results at their current form. 

We acknowledge the reviewers concern and appreciate the reviewer’s viewpoint on the PTM-based analysis presented in our study. We recognize the challenges and limitations inherent to our study regarding the analysis of post-translational modifications (PTMs), especially concerning the functional relevance of specific modification sites on proteins and the dynamic nature of PTMs. Within its limitations, our study aimed to be a pioneer snapshot view into the landscape of PTMs, specifically acetylation and glycation, within gut and plasma EVs. Moreover, our approach represents an initial expedition into looking at PTM modifications present within EVs proteome.

While we acknowledge the limitation in fully capturing the dynamic range of PTMs due to the snapshot nature of our analysis, our methodology was designed to provide a comprehensive overview of the PTM landscape within EVs under different dietary conditions. We indeed detected multiple PTMs on the same protein species. We have carefully considered the differences in PTM patterns between normal chow diet (NCD) and high-fat diet (HFD) samples. The distinct PTM profiles observed suggest that the modifications we detected are not merely artifacts of sample preparation denaturation stress but reflect biologically relevant differences influenced by the dietary conditions. This observation emphasizes the potential significance of our findings, indicating that PTMs within EVs may indeed mirror changes in the cellular and systemic environment induced by obesity. In light of these considerations, we have further elaborated on these points in the discussion section of our manuscript, acknowledging the limitations while also highlighting the potential of our findings to shed light on the role of PTMs in EV-mediated communication and the pathophysiology of obesity. We believe that our study lays the groundwork for future research that can investigate deeper into the functional implications of specific PTMs and their sites, thereby enhancing our understanding of EV biology in health and disease.

Reviewer 3 Report

Comments and Suggestions for Authors

The study of Pereira et al. provides valuable information on the changes in protein profile of the extracellular vesicles resulting from high-fat diet and relevance for obesity and metabolic disorders. I would recommend only minor revisions in regards to Figure quality. It would be nice to have a graphical abstract that summarizes the major findings of these proteomics analyses.

Comments on the Quality of English Language

The English style and grammar are clear and need only minor revision for a few typos.

Author Response

We sincerely appreciate your thoughtful and constructive feedback regarding our manuscript. We are especially grateful for your suggestion to include a graphical abstract, which we believe will neatly communicate our major findings. In response to your recommendations, we have carefully revised the manuscript to address minor English language corrections throughout the text. Additionally, we have corrected the few typos errors.

Round 2

Reviewer 1 Report

Comments and Suggestions for Authors

Most of my concerns were well addressed.

Comments on the Quality of English Language

Acceptable